# A VetCompass Australia Study of Antimicrobial Use in Dog-to-Dog Bite Wounds (1998–2018)

**DOI:** 10.3390/antibiotics11010055

**Published:** 2022-01-02

**Authors:** Nicole Jacqueline Kalnins, Catriona Croton, Mark Haworth, Justine Gibson, Sarah Leonie Purcell, Allison Jean Stewart

**Affiliations:** 1School of Veterinary Science, The University of Queensland, Gatton, QLD 4343, Australia; n.kalnins@uq.edu.au (N.J.K.); m.haworth@uq.edu.au (M.H.); gibson.j@uq.edu.au (J.G.); sarah.purcell@uq.edu.au (S.L.P.); 2Faculty of Health, Engineering and Sciences, School of Sciences, University of Southern Queensland, Toowoomba, QLD 4350, Australia; catriona.croton@usq.edu.au

**Keywords:** bite wounds, canine, antimicrobial susceptibility, antimicrobial stewardship, bacteriology, VetCompass Australia

## Abstract

Although dog-to-dog bite wounds (DBW) are a common presentation to veterinary clinics, antimicrobial prescribing habits of Australian clinics have not been reported. This study determined the frequency and results of DBW cultures; antimicrobial selection; and importance class of antimicrobials prescribed relative to wound severity, geographic location, or year. A systematic sample of 72,507 patient records was retrieved from the VetCompass Australia database. Records for 1713 dog bite events involving 1655 dogs were reviewed for presenting signs, results of culture and susceptibility testing (C&S), antimicrobial treatment, geographical location, and outcome. A crossed random effects multivariable logistic regression model was used to determine if antimicrobial importance was associated with wound severity, year, and location, and to assess the differences in antimicrobial prescription between geographical locations, clinics, and veterinarians. Antimicrobials were prescribed in 86.1% of DBW. Amoxicillin-clavulanic acid was prescribed in 70% (1202/1713) with underdosing in 15.8% (191/1202). High-importance antimicrobial use was associated with wound severity (*p* < 0.001), year category (*p* = 0.007), and surgery (*p* = 0.03). C&S testing was recorded as having been performed in only one case. Differences in individual veterinarian prescribing habits were stronger than the clinic culture, suggesting that education utilizing clinic-wide antimicrobial guidelines may aid in improving antimicrobial stewardship.

## 1. Introduction

The VetCompass Australia (VCA) program is a veterinary medical record-based research platform that collects and aggregates de-identified clinical records from both primary care and specialist Australian veterinary practices. It is based on the VetCompass United Kingdom model, which has been collecting data since 2010 and is a leading source of veterinary epidemiological data [1]. Since 2016, 181 practices across Australia have contributed data to the VCA program, which represents 5.6% of Australian veterinary clinics [2]. It is the largest companion animal clinical data repository of its kind in Australia and can be readily queried, enabling large scale extraction of antimicrobial usage patterns of companion animal practices throughout Australia [1,2,3]. Analysis of VCA clinical records has been used to identify geographical and temporal trends in the prevalence of inherited and acquired diseases, therapeutic analysis, and as an aid in clinical auditing [1]. The platform thus provides sustainable and cost-effective access to veterinary data across Australia, with the goal of achieving an improved level of patient care for companion animals through data analysis [1].

In order to improve antimicrobial prescription practices, it is important to have an understanding of how antimicrobials are currently being used. Comparisons can then be made with antimicrobial guidelines, highlighting where further education is required [4]. Data on antimicrobial use of individual clinicians can be used to help educate and provide feedback to clinicians on their prescribing practices. This type of auditing has been shown to improve patient outcomes and decrease the rate of unnecessary antimicrobial prescription in human medicine [4,5]. To date, there have been no published studies on the monitoring or auditing of antimicrobial prescription practices of veterinarians throughout Australia [6].

Dog-to-dog bite wounds (DBW) are a common presentation to veterinary clinics often requiring antimicrobial therapy. Currently, there is no published data on the most commonly prescribed antimicrobials for the treatment of DBW across Australia or the most commonly cultured pathogens. It is recommended that all infected DBW are cultured to determine the most appropriate antimicrobial selection, however, culturing is rarely performed [7]. University veterinarians in Queensland, Australia, were recently found to be prescribing antimicrobials in 88.1% of dog-to-dog bite wounds (DBW) [8]. Amoxicillin–clavulanic acid was prescribed in 73.4% of 1526 dogs, however, underdosing occurred 13% of the time, while first-generation cephalosporins were prescribed in 18.1% and underdosed 26% of the time. Increased use of high-importance antimicrobials was associated with wound severity, antimicrobial polytherapy, and year. High importance antimicrobials were used less frequently in clinics with specialists, but culture and susceptibility testing (C&S) was only performed in 1.8% of DBW. Further education of university-employed veterinarians was recommended to optimize antimicrobial stewardship [8]. It is unknown if the results from this study would reflect the national antimicrobial prescribing habits of Australian veterinarians. VetCompass Australia data is an untapped resource to help assess antimicrobial stewardship in DBW in Australia and determine if education is required to improve prescription practices for veterinarians.

The objectives of this study were to: 1. determine the most common antimicrobials prescribed to treat DBW in Australia and if wound severity influenced the prescription of the importance class of antimicrobials used; 2. Identify any variability in the importance class of antimicrobial prescribed between geographical location, clinic, and veterinarian; 3. Report underdosing of antimicrobials; and 4. Report the occurrence of C&S testing and identify the most prevalent organisms cultured and their antibiogram.

## 2. Results

A total number of 72,507 electronic patient records (EPRs) were identified from the VCA database. Patients with incomplete signalment (*n* = 2624) were excluded, leaving 69,883 patients. A systematic sampling strategy was used to select 3196 patients, of these, 1483 records were excluded due to being incomplete (*n* = 301); the presenting complaint was a medical condition other than a DBW (*n* = 1064), or unknown trauma (*n* = 118). The final dataset for analysis consisted of 1713 unique DBW events, and 1655 dogs were included in this analysis (Figure 5). These dogs were seen by 644 veterinarians employed at 123 clinics. The dogs had up to four unique DBW events, with 97% (*n* = 1605) of dogs having one event, 2.6% (*n* = 43) of dogs two events, 0.4% (*n* = 6) three events, and 0.1% (*n* = 1) four events. Each veterinarian examined 1–17 unique DBW consults, and the clinics saw 1–91 unique DBW events. There were up to 25 veterinarians recorded at each clinic, and up to six clinics per veterinarian with 18.0% (116/644) of veterinarians recorded at more than one clinic, indicating that the veterinarians crossed between clinics. All dogs and clinics had a unique de-identified code, however, there were 168 consultations missing a veterinarian code. The consultations in the sample were conducted between June 1998 and August 2018 and were from Queensland (QLD), New South Wales (NSW), Victoria (VIC), South Australia (SA), Australian Capital Territory (ACT), and Western Australia (WA).

### 2.1. Signalment

The study population of 1655 dogs consisted of 901 (54.4%) males, of which 547 (60.7%) were desexed; and 754 (45.6%) females, of which 578 (76.7%) were desexed. The median age was 4.6 years old (interquartile range [IQR]: 2.0 to 8.0) with a range of 1 month to 19.6 years). The median weight was 19 kg (IQR: 8.4 to 26.2) with a range of 2.6–69.2 kg (Table 1). There were 993 purebreds (60.0%) and 662 crossbreds (40.0%). Of the pure breeds, 34.7% were terriers (*n* = 345), 15.9% working dogs (*n* = 157), 11.0% toy breeds (*n* = 109), 10.9% utility dogs (*n* = 108), 10.9% non-sporting dogs (*n* = 108), 9.7% gun dogs (*n* = 97), and 6.9% hounds (*n* = 69) [9].

### 2.2. Antimicrobials

At least one oral or parenteral antimicrobial was prescribed for 86.6% (1483/1713) of events with 4.2% (72/1713) received topical antimicrobials only, and 9.2% (158/1713) received no antimicrobials. When antimicrobials were not prescribed, they were deemed unnecessary by the primary veterinarian in 91.7% (145/158) of events; in 3.9% (6/158) of events the owners declined antimicrobials, and eight dogs either died or were euthanised prior to antimicrobial administration. The majority, 51.8% (82/158) of these events, were categorised as grade one wounds. Of the dogs that either died or were euthanised prior to receiving antimicrobials, three dogs had grade four wounds and five dogs had grade five wounds.

Of the 1555 DBW events in the full dataset where oral or parenteral antimicrobials were prescribed, 83.3% (*n* = 1296) received monotherapy (one antimicrobial prescribed, which may have been in different formulations such as injectable and oral) and 16.6% (*n* = 259) received polytherapy (greater than one antimicrobial prescribed) (Table 1). When dogs received polytherapy, 64.8% (168/259) were prescribed two antimicrobials, 4.2% (11/259) were prescribed three antimicrobials, and 0.4% (1/259) were prescribed four antimicrobials. The most commonly prescribed antimicrobials used in polytherapy were amoxicillin-clavulanic acid, fluoroquinolones, first-generation cephalosporins, and metronidazole (Figure 1). Low importance antimicrobials were prescribed in 7.3% (126/1713) of events, 79.4% (1360/1713) of events were prescribed as medium importance antimicrobials, and 4.0% (69/1713) were prescribed as high importance antimicrobials.

Amoxicillin–clavulanic acid was the most commonly prescribed antimicrobial with 70% (1202/1713) of events receiving parenteral and/or oral formulations. Thirty seven percent (631/1713) received just the oral formulation (tablet or liquid), 5.1% (88/1713) received a parenteral injection only, and 28.1% (484/1713) received both. First-generation cephalosporins were the second most frequently prescribed antimicrobial with 11.5% (197/1713) of events receiving parenteral (cefazolin) and/or oral (cefalexin) formulations. Nine percent of events (159/1713) received a single oral dose, 0.7% (13/1713) received a single parenteral injection, and 1.4% (25/1713) received both formulations. Third-generation cephalosporins (cefovecin) were prescribed in 1.5% (27/1713) of events and were prescribed as monotherapy for 96% of these cases. Enrofloxacin was prescribed in 2.4% (42/1713) of events, 80% of which was prescribed as a part of polytherapy. Metronidazole was prescribed in 5.1% (88/1713) of events and lincosamides in 1.4% (24/1713). A topical antimicrobial was prescribed in 8.5% (147/1713) of events, with 4.2% (72/1713) having only a topical antimicrobial prescribed (Table 2). Antimicrobial dosage, frequency, and duration of administration for the full dataset is shown in Table 3.

Prescribed dosages were compared to recommended dosages [10,11] and were able to be determined in 1389/1713 events. Events that did not receive antimicrobials, received only topical antimicrobials, or were missing information on dosage, were excluded from this analysis. At least one inappropriately low dose was prescribed in 33.6% (467/1389) of events. Appropriate dosages were prescribed in 41.9% (583/1389) of events and 24.4% (339/1389) were prescribed at least one high dose. Inappropriately low doses of low importance antimicrobials were administered in 29.3% (37/126) of events. Twenty-nine percent (400/1360) of medium importance antimicrobials were underdosed and 43.4% (30/69) of high importance antimicrobials were underdosed (Table 4). Both oral and parenteral amoxicillin-clavulanic acid were prescribed at dosages less than 12.5 mg/kg in 15.8% (191/1202) of prescriptions. Oral and parenteral first-generation cephalosporins were prescribed at dosages less than 22 mg/kg in 44.1% (87/197) of prescriptions and third-generation cephalosporins were prescribed lower than 8 mg/kg in 51.8% (14/27). Enrofloxacin was prescribed at lower than the recommended 5 mg/kg dose in 11.9% (5/42) of prescriptions.

Out of the 1713 dog bite events, only one (0.1%) C&S test was reported, and the results were not recorded in the EPR.

### 2.3. Year Category

The proportion of DBW events presented for treatment between the time periods of 1998–2004 was 18.7% (321/1713), 29.3% (501/1713) for 2005–2009, 30.2% (518/1713) for 2010–2014, and 21% (373/1713) for 2015–2018 (Table 1).

There was an association between antimicrobial importance (including no antimicrobials prescribed) and year category (χ^2^ (9) = 18.3, *p* = 0.03). For the time periods of 1998–2000 and 2015–2018, there was an increase (9.0% to 11.5%) in the number of consults where no antimicrobials were dispensed (Figure 2). There was a concurrent decrease in the proportion of consultations where low importance antimicrobials were dispensed from 11.8% to 5.4% and an increase in dispensing of medium importance antimicrobials from 75.4% to 78.3%, and in high importance antimicrobials from 3.7% to 4.8%.

### 2.4. Geographic Location

The majority of consultations for DBW events occurred in QLD 58% (993/1713), followed by NSW at 22.3% (382/1713) and VIC at 13.1% (224/1713). South Australia had 3.6% (62/1713) of DBW events, the ACT 2.9% (49/1713), and WA only had 0.2% (3/1713) (Table 1). There was no association between state (excluding WA due to low numbers) and antimicrobial importance of drugs dispensed in this dataset (χ^2^ (12) = 18.3, *p* = 0.11) (Figure 3).

### 2.5. Wound Grade

Of the 1713 DBW events, 12.8% (*n* = 219) sustained grade one wounds, 13.0% (*n* = 223) grade two wounds, and 26.7% (*n* = 457) grade three wounds. The majority of dogs, 46.6%, (*n* = 799) sustained grade four wounds and only 0.9% (*n* = 15) of dogs sustained grade five wounds. There was a strong association between wound severity grade and antimicrobial importance class prescribed in the full dataset (*p* < 0.001) (Figure 4).

### 2.6. Time of Injury to Presentation

The time from injury to presentation for treatment was recorded for 64.8% (1111/1713) of events. The majority of events, 55.9% (622/1111), presented >24 h post injury. Six percent (68/1111) presented within 1 hour, 16.8% (187/1111) between 1–8 hours post injury, and 21.0% (234/1111) between 8–24 h post injury. Wound grades were compared in these 1111 events and a strong association was found between the time of injury to presentation and wound severity (χ^2^ (16) = 46.6, *p* = <0.001). The majority of grade one to four wounds presented >24 h post injury: 66.4% (85/128) of grade one, 45.8% (66/144) of grade two, 47.7% (139/291) of grade three and 63.6% (331/520) of grade four wounds. Sixty-two percent (5/8), of grade five wounds presented between 1–8 hours post injury.

### 2.7. Surgery

Surgery was performed in 25.4% (435/1713) of the DBW events. In 71.4% (1224/1173) of the events, surgery was deemed unnecessary by the primary care veterinarian, and in 3.2% (54/1713) surgery was recommended but declined by the owner. Of the 457 grade three wounds, 13.3% (61/457) underwent surgery, in 83.8% (383/457) surgery was deemed unnecessary, and in 2.8% (13/457) the owners declined surgery. Of the 799 grade four wounds, 40.8% (326/799) underwent surgery, in 55.3% (442/799) surgery was deemed unnecessary, and in 3.9% (31/799) the owners declined surgery. There were 15 grade five wounds and 40.0% (6/15) underwent surgery. Nine dogs in this category did not undergo surgery and were euthanised. Surgery was not required in any grade one or two wounds. There was a strong association between surgery being performed and the importance of antimicrobials prescribed (χ^2^ (6) = 60.1, *p* < 0.001).

### 2.8. Hospitalisation

The majority of dogs were treated as out-patients and spent less than 24 h in hospital (75.1%, 1288/1713). Hospitalisation was measured in days, and the median time of hospitalisation was 0 days. The maximum time of hospitalisation was 4 days. Ninety-eight percent (216/219) of grade one wounds, 84.3% (188/223) of grade two wounds, 87.7% (401/457) of grade three wounds, and 59.5% (476/799) of grade four wounds were treated as out-patients. Forty-six percent (7/15) of grade five wounds spent less than 24 h in hospital, however, this was due to euthanasia, not discharge home.

### 2.9. Re-Examination, Complications, Mortality

Re-examination was only performed in 14.1% (243/1713) of DBW events, 84.7% (1451/1713) were lost to follow up, and 1.1% (19/1713) were referred back to their primary care veterinarian. In events that presented for re-examination, 90.5% (220/243) had no complications and 9.4% (23/243) had recorded complications, 6.5% (16/243) of which were consistent with possible infection. Complications included: inflammation of the site (43.7%, *n* = 7), seroma formation (37.5%, *n* = 6), purulent discharge (25%, *n* = 4), wound dehiscence (12.5%, *n* = 2), and abscessation (12.5%, *n* = 2). The majority of complications occurred in grade four wounds (75%, 12/16). Thirty-seven percent (*n* = 6) of dogs with grade three wounds, 18.7% (*n* = 3) with grade two wounds, and 12.5% (*n* = 2) with grade one wounds had complications consistent with possible infection. No grade five wounds had complications consistent with possible infection.

The mortality rate as a direct result of the DBW was 1.7% (29/1713). Twenty dogs were euthanised either due to financial reasons or the extent of their injuries, and nine dogs underwent cardiovascular arrest. No dogs with grade one or two wounds arrested nor were euthanised. Of the dogs which were euthanised, 1 had grade three wounds, 11 had grade four wounds, and 9 had grade five wounds. Of the dogs that arrested, two had grade three wounds and seven had grade four wounds. No dogs with grade five wounds underwent cardiopulmonary arrest and died.

### 2.10. Model for Association between Higher Importance Antimicrobials and Wound Severity, State and Year

An additive crossed random effects ordinal logistic regression model was used to assess associations between the use of higher importance antimicrobials with the explanatory variables wound severity, state, and year of primary interest. These variables were trialed as fixed effects (Table 5), with state removed from the model, as it was not significant (*p* = 0.94). For the explanatory variables of secondary interest, recommendation or performance of surgery was found to be significant as a fixed effect in the model and was therefore included. The remaining variables of patient age, sex, neuter status, weight and breed category; clinic type (general practice or specialist); length of hospitalisation; season of attack; and time of injury to presentation were not statistically significant and were removed. Given the low numbers of C&S testing, this was not trialed as a variable.

After multivariate adjustment, events with a higher wound severity were associated with higher importance antimicrobials dispensed, with an estimated increase in odds of 75% (95% confidence interval (CI): 43 to 113) per wound severity. Year category was associated with importance of antimicrobials dispensed (*p* = 0.007), with a significant increase in odds of a higher importance antimicrobial being dispensed between the year categories of 2010–2014 and 2015–2018, compared with the baseline of 1998–2004. There was also an increase in the estimated odds between the years of 2015–2018 and 2005–2009 of 90% (95% CI: 8 to 234%, *p* = 0.03).

The random effects for both clinic and veterinarian contributed significantly to the model (*p* < 0.001 for both); the importance of the chosen antimicrobial varied significantly between clinics and veterinarians. The random effect at the veterinarian level had an estimated variance of 2.14 (95% CI: 0.95 to 4.82), which was larger than that of the clinic at 0.54 (95% CI: 0.21 to 1.38) (Table 5). Although both were statistically significant and the point estimate of the effect of clinic on importance of antimicrobial prescribed was smaller, the confidence intervals overlapped. A random effect for dogs was trialed and removed as it did not contribute significantly to the model (*p* = 1.0). A likelihood-ratio test showed sufficient variability between clinics and veterinarians for the mixed-effects ordered logistic regression model to be preferred over the standard ordered logistic regression (χ^2^ (2) = 33.6, *p* < 0.001).

## 3. Discussion

Empirical antimicrobial therapy was administered to 86.6% of dogs presenting for treatment of DBW. Medium importance antimicrobials amoxicillin–clavulanic acid and first-generation cephalosporins were the most commonly prescribed, however, they were prescribed below the recommended dose range [10,11] in 15.8% and 44.1% of dogs, respectively. At least one antimicrobial was prescribed at an inappropriately low dose in 33.6% of DBW events. Concerningly, 43.4% of high importance antimicrobials were prescribed at lower than the recommended dose. There was increased use of antimicrobials of high importance with increasing wound severity, surgical intervention, and years. However, over the time period studied, there was an increase in the number of cases in which veterinarians deemed antimicrobials unnecessary. No difference was found in the prescription of antimicrobial importance class between Australian states. The variation in antimicrobial prescription was greater between individual veterinarians within the same clinic than when compared with between-clinic prescription. Although one of the initial aims of the study was to the determine the occurrence of C&S testing in DBW and identify the most commonly cultured bacteria, only one C&S test was recorded as having been performed in the records manually reviewed, and no results were available.

The most common antimicrobials prescribed for the treatment of DBW by Australian veterinarians in this study were similar to a recent retrospective of three QLD-based university teaching hospitals (UQVETS) [8]. Amoxicillin–clavulanic acid was prescribed as antimicrobial to 70.2% of dogs with DBW in this study compared to 73.4% of dogs in the UQVETS study. First-generation cephalosporins were the second most frequently prescribed antimicrobial for DBW with 11.5% and 18.1% prescribed by veterinarians across Australia and UQVETS, respectively. Third-generation cephalosporins were prescribed at a slightly higher rate to DBW throughout Australia (1.5%) compared to UQVETS (0.8%). However, enrofloxacin was prescribed at a higher rate by veterinarians at UQVETS (7.6%) compared to Australian veterinarians (2.4%). This may be due to clinic type, as two of the three UQVETS hospitals were predominantly emergency and specialist referral hospitals, whereas the majority of the VCA clinics analyzed were general practice.

Another VCA study that assessed antimicrobial usage patterns for all medical conditions in companion animal veterinarians also reported amoxicillin–clavulanic acid was the most commonly prescribed antimicrobial, being given to 34% of dogs requiring antimicrobials [2]. Similar to this study, they found the two most commonly prescribed antimicrobials of high importance were third-generation cephalosporins and fluoroquinolones, prescribed to 3.1% and 3.5% of dogs, respectively. Two recent studies have assessed the prescription behaviors of Australian veterinarians [6,7]. Both studies found amoxicillin–clavulanic acid was the most frequently prescribed antimicrobial for any medical or prophylactic treatment followed by first or second-generation cephalosporins. Fluoroquinolones were prescribed in 18% of canine cases in one study [7], whilst the second study reported enrofloxacin was ‘sometimes’ prescribed and third-generation cephalosporins were ‘rarely’ prescribed; however, 49.3% of the respondents indicated they were uncomfortable prescribing either fluoroquinolones or third generation cephalosporins [6]. Reasons for discomfort included the importance rating of these antimicrobials, the requirement to perform C&S testing with their use, and concerns regarding the promotion of antimicrobial resistance [6].

The antimicrobial prescribing practices reported in these Australian studies are similar to what has been reported globally. A survey of New Zealand veterinarians found amoxicillin–clavulanic acid was prescribed in 48% of cases (canine and feline) followed by first-generation cephalosporins at 31% and fluoroquinolones at 11% [12]. Two studies assessing antimicrobial prescription of companion animal veterinarians across the United Kingdom, (one of which used the VetCompass UK database) found amoxicillin–clavulanic acid was the most common first-choice antimicrobial prescribed to 28% and 45% of dogs who were treated with an antimicrobial agent [13,14]. Both studies also found fluoroquinolones were the most commonly prescribed antimicrobial of high importance to dogs. Across Belgium, Italy, and The Netherlands, amoxicillin–clavulanic acid was also the first-choice antimicrobial, prescribed to 27% of dogs who were treated with an antimicrobial agent [15]. In Norway, amoxicillin–clavulanic acid is again the most common antimicrobial prescribed, and the authors found its prescription increased over their study period of 2004–2008 from 26.5% to 43.7%. They also found a 23.6% increase in the prescription rate of fluoroquinolones during the study period [16]. Amoxicillin–clavulanic acid was also the most frequent first choice for veterinarians in Chile, followed by enrofloxacin as the most common secondary antimicrobial when polytherapy was implemented [17]. The high level of empirical use of this β-lactam both in Australia and globally considerably increases the risk of development of β-lactamase resistance in bacteria and can select for resistance to aminoglycosides and fluoroquinolones [14]. It also increases the risk of horizontal transmission of these resistant bacteria to the human population [14,17].

An increase in antimicrobial importance prescription was associated with increasing wound severity, surgical intervention, and year category. In this study, there was a 75% increase in the odds of a higher antimicrobial importance class being prescribed for each increase in wound severity by one grade. This study also found that dogs who underwent surgery as part of wound treatment had a 75% increase in the odds of a higher antimicrobial class being prescribed. Infection of more severe wounds (grade four and five) is likely to be life threatening as they may penetrate into the thoracic or abdominal cavity or synovial structures and often require surgical debridement and reconstruction. Clinicians would be more likely to use broad spectrum antimicrobials and polytherapy to achieve four quadrant coverage in these cases, as no single antimicrobial is effective against all species of bacteria, which are commonly cultured from DBW [18]. Antimicrobial stewardship guidelines recommend de-escalation of antimicrobials of high importance and polytherapy based on the results of C&S testing and clinical improvement [19,20]. In this study, the C&S testing rate was 0.1%, so if de-escalation was being performed in these cases, we would assume it is based on clinical improvement.

This study found a marked increase in the use of high importance antimicrobials from 1998–2004 to 2010–2014 and 2015–2018. There was also a significant increase from 2005–2009 to 2015–2018. This is similar to the study assessing prescription trends in our UQVETS study, which found a 5.7 times increase in the odds ratio of prescribing an antimicrobial of high importance from 1998–2004 to 2010–2014 [8]. This increase may be due to registration of enrofloxacin and cefovecin in Australia for veterinary medicine in 1995 and 2008 respectively. The earliest introduction of antimicrobial stewardship guidelines for dogs, specifically for DBW in Australia, was in 2013 [19]. If veterinarians were closely following the guidelines, a reduction in prescription of high importance antimicrobials from 2015–2018 would be expected. From 1998–2000 to 2015–2018, there was, however, a small increase in the percentage of consultations where no antimicrobials were prescribed, possibly indicating a decrease in the prescription of prophylactic antimicrobials.

This study found no association between state and the importance category of antimicrobials prescribed for the treatment of DBW. An effect may have been detected in a larger sample size and if Tasmania and the Northern Territory were represented. The majority of clinics were also located in urban regions, so a comparison between urban and rural antimicrobial prescription was unable to be made. A study assessing antimicrobial usage patterns in Australian companion animal veterinarians using VCA data, found a difference in the prescription rate of antimicrobials between states [2]. The authors assessed 61 clinics in QLD, 35 clinics in VIC, 31 clinics in NSW and the ACT, 6 clinics in SA, and 4 clinics in WA. It was found that SA, NSW and ACT, and VIC had the highest rate of prescription with a median of 140 consultations dispensing antimicrobials per 1000 consultations. This was followed by QLD and WA with 129 and 123 consultations dispensing antimicrobials per 1000 consultations. There was also a difference in the rate of prescription of high importance antimicrobials between states. Western Australia had the highest rate of prescription of high importance antimicrobials with 49 consults per 1000 prescribing an antimicrobial of high importance. Queensland, NSW, ACT, and SA had a prescription rate of 40 consults per 1000 prescribing an antimicrobial of high importance, and VIC was marginally lower at 39 consults per 1000. These authors also found antimicrobial use was lower in inner and outer regional areas compared to major cities, but the use of high importance antimicrobials was higher in outer regional areas and major cities compared to inner regional areas [2]. These authors analysed 595,089 consultations where antimicrobials were prescribed to dogs and cats. It is possible our study may have detected differences between states if a similar sample size was analysed.

Spatial antimicrobial prescription patterns have been assessed in companion animal veterinarians in Norway between 2004–2008 and the UK between 2012–2014 [14,16]. Regional prescription differences were noted in Norway, with higher prescription rates in larger towns attributed to a higher density of dogs and number of clinics within the areas [16]. When assessing specific antimicrobials, they found higher prescription rates of penicillin, tetracyclines, trimethoprim sulphonamide, and aminoglycosides in rural districts. Overall, they found an increase in prescription rates, including fluroquinolones, across the country over the study period. The authors suggested that the regional differences in prescription behaviors may be due to differing prescription attitudes among veterinarians and clients in cities compared to rural areas, as well as social and economic factors [16]. The study of UK spatial antimicrobial prescription patterns used the VetCompass UK database and found 25% of dogs attending a veterinary practice received at least one antimicrobial over their study period [14]. They found a greater rate of antimicrobial prescription in the southeast England, south Wales, and the southwest of Scotland. Reasons for this geographical variation may include differences in animal demographics and diseases, as well as prescription behaviors of veterinary practices, however, the authors state the true underlying reasons for the regional differences need to be investigated further. This study also excluded emergency, referral, and charity-based hospitals from their data and, therefore, antimicrobial prescription rates may differ when these are taken into account [14]. Knowledge of geographical patterns of antimicrobial usage, especially antimicrobials of high importance, is essential for the implementation of risk-based monitoring strategies for antimicrobial resistance. Areas with higher rates of prescription, especially of high importance antimicrobials, should be targeted for the monitoring of antimicrobial-resistant bacteria. Antimicrobial stewardship campaigns for prudent use can also target veterinarians in those regions, hopefully reducing the prescription of antimicrobials of high importance [16]. As our study did not show differences between the Australian states, then education relating to antimicrobial prescription guidelines can continue to be at a national rather than a local level.

The importance of the chosen antimicrobial varied significantly between clinics and veterinarians in this study, with the point estimate of the variance component for the clinic being lower than the veterinarian. This suggests antimicrobial prescription by clinics varies less than between individual veterinarians, indicating less of a clinic culture and more independence from veterinarians. However, given that the confidence intervals of the variance components overlapped, a clear indication of difference in the variance components cannot be inferred. In contrast, human medicine has demonstrated a strong influence of clinic culture on prescribing patterns. Medical hierarchy and professional relationships were considered a major factor, with senior colleagues having significant influence over prescription behavior of junior colleagues. Senior doctors were also less likely to follow policy and guidelines, relying on personal knowledge and experience. To influence prescription behaviour of individuals, interventions are required that address clinical leadership and peer approval within existing groups to change prescription behavior not just in the implementation of policies and guidelines [5].

The variation between individual veterinarians’ choice of antimicrobial prescription may be influenced by personal experience, the university they undertook their veterinary degree, year of graduation, mentoring by senior veterinarians within a practice, recent continuing education, and/or literature review. The VCA data is de-identified, so this study had no access to specific veterinarian demographics. However, one recent study found no difference between year of graduation and the use of high importance antimicrobials in Australian veterinarians [7]. Several surveys found that veterinarians identified that the most importance sources that influence their prescription behaviors were information from published literature and training, label and package information, results of antibiograms, and colleagues’ and personal experience [6,21,22]. One study found clinic culture was one of the least important influencing factors [23].

A study evaluating VCA data to assess antimicrobial usage patterns of Australian companion animal veterinarians found a variation in the prescription rate of antimicrobials between individual clinics as well as the prescription of high importance antimicrobials [2]. Specifically, they found emergency and referral centers had a higher rate of both overall antimicrobial prescription and the prescription of antimicrobials of high importance compared to general practice. They hypothesized the difference in antimicrobial prescription between clinic types could be due to the predominant consultation type, as routine examinations and vaccinations do not frequently require antimicrobials [2]. Although this current study found differing results, it only included a limited number of referral and emergency clinics.

This study identified 41.9% of prescriptions that followed recommended antimicrobial prescription guidelines, but of concern, 33.6% of DBW events were underdosed. Of the DBW cases prescribed amoxicillin–clavulanic acid, 15.8% were underdosed. Even more concerning is the overall rate of antimicrobial underdosing of 33.6% of DBW across Australia. By gaining an understanding of which groups of veterinarians are inappropriately prescribing and underdosing empirical antimicrobials for DBW, targeted education programs through regulating bodies such as the Australian Veterinary Association and nation-wide surveillance for bacterial resistance can be implemented more effectively. Though 4% of DBW events were prescribed an antimicrobial of high importance, the reported rate of C&S testing was extremely low at 0.5%. It was out of the scope of this study to assess why the rate of C&S testing was so low, however, it may be due to the perceived increased costs to the client. This has previously been described as the main reason to not perform C&S testing by Australian veterinarians, with many opting for treatment trials with empirical antimicrobials [23]. Identifying the reasons underlying the lack of C&S testing can help guide education programs and increase antimicrobial stewardship. Obtaining wound cultures for organism identification and antimicrobial susceptibility testing is considered the gold standard for the treatment of any wound, not just DBW. Culture and susceptibility testing should ideally be considered routine practice for any wound, not only reserved for severe cases or treatment failure [24,25]. Submission of appropriate specimens for bacterial culture, pathogen identification, and susceptibility testing allows for an evidence-based approach for drug selection [26]. It can, however, increase the costs associated with patient care, although other costs associated with improper drug selection (prolonged treatment, use of expensive antimicrobials, and increased morbidity and mortality) can outweigh the financial costs of testing [26]. A recent study found that amoxicillin–clavulanic acid was an appropriate first-line treatment of grade three and four DBW and empirical treatment with high importance antimicrobials, specifically enrofloxacin was not warranted in 96% of cases. The use of C&S testing prior to commencing antimicrobial therapy was able to identify the rare cases that were not successfully treated with amoxicillin–clavulanic acid [27].

The extent to which antimicrobial resistance is affecting the health of humans and animals is not fully known. There are concerns that emerging resistance among bacteria could escalate, with unpredictable efficacy of antimicrobials, and some bacterial infections could become untreatable [26]. There is now increasing global pressure to develop strategies to protect the effectiveness of existing and new antimicrobials by reducing selection pressure, driving emerging resistance in bacteria [26,28]. Methicillin-resistant *Staphylococcus aureus* (MRSA) is a significant human pathogen in hospitals and communities. MRSA is now recognized as an emerging pathogen of animals. Resistant infections might result from treatment with inappropriate antimicrobials [29]. As these pathogens are considered to be transmissible between humans and animals, they pose a genuine threat not only to hospitalized animal patients but also to the work place safety of veterinary personnel and animal owners [30]. Veterinarians should therefore critically evaluate current antimicrobial usage to identify ways to prevent MRSA infections such as hand washing, masks, and use of topical disinfectants in an attempt to improve the health and wellbeing of animals and humans.

The main limitation of this study is its retrospective nature. The accuracy and reliability of the results depends on the completeness and quality of the data recorded within the medical records. Misclassification bias may exist as the data collected was limited to the data available in the medical records. Misclassification and confounding biases may have also resulted from the lack of uniformity between the large number of clinics and clinicians in their diagnostic criteria, clinical management, and record keeping. Wound severity grades were determined based on interpretation of wound description in the clinical record, which may have also led to misclassification. Missing data from the medical records could have resulted in selection bias, especially in cases where re-examination and follow up records were inaccessible. Selection bias could have also resulted from the spatial distribution of veterinary clinics that have signed-on to the VCA collection database. This effect could be countered by broadening the base of data contributors in under-represented areas. Inclusion of a wider variety of clinic types (emergency, specialist, rural and urban) and locations would assist in overcoming the demographic biases that may arise [1].

## 4. Materials and Methods

### 4.1. Data Source and Search

De-identified medical records were extracted from the VetCompass Australia database (version 0.3) for dogs with a possible DBW. Electronic patient records (EPRs) were identified using a Structured Query Language search for the expressions: DFW, DBW, dog fight, dog attack, dog bite in the examination text field, and “canine” in the species field. Consultations taking place within 2 months of the identified DBW consult were also extracted to determine if re-examinations were performed, in case the EPR lacked the relevant DBW expressions. Human ethics for the VetCompass Australia data was obtained.

### 4.2. Data Clean

A total of 123,186 unique consults were extracted, containing 72,507 patients from 164 clinics. As there was a large number of cases, a systematic sampling strategy analyzing every 10^th^ record was used to allocate cases for evaluation with the starting point at the first consultation recorded and a total of two passes performed. Manual evaluation of 3196 consults was performed. Electronic patient records were retained in the study if the cause of injury was a known DBW as determined from the examination text field. The EPRs were excluded if the injuries were not definitive for a DBW (e.g., found with wounds with no witnessed dog attack or presence of other dogs) or EPRs were incomplete (Figure 5). The data was exported to Microsoft Excel (version 16.43) for manual labelling. Information collected included general patient signalment (age, sex, intact status, breed, and weight), year of consultation, time from injury to presentation, wound severity, antimicrobials prescribed (dose, frequency and duration), culture and susceptibilities performed and their results, treatment outcomes (duration of hospitalisation, rechecks, complications, and mortality directly associated with the injuries), state, clinic, and veterinarian. Pure breeds were categorized based on the Australian National Kennel Council Ltd. [9]. All dogs and clinics had a unique de-identified code, however, there were 168 consultations missing a veterinarian code.

A previously established grading system was used to categorize wound severity using the examination text [31,32,33]. Grade one and two wounds were categorised as superficial wounds with partial thickness and full thickness laceration of the dermis, respectively. Grade three wounds were full thickness puncture wounds with penetration of the dermis without systemic illness. Grade four wounds were full thickness punctures or lacerations with avulsion of underlying tissues and dead space, underlying muscle trauma, possible penetration of a joint, abscess, and/or systemic illness. Grade five wounds were severe and included penetration into body cavities (abdomen, thorax) and open fractures. A laceration was defined as a wound >10 mm in length and a puncture as a wound <10 mm in length.

Antimicrobials dispensed were classified as low, medium, and high importance as defined by the Australian Strategic and Technical Advisory Group (ASTAG) of antimicrobial resistance (Table 6) [34]. Dose rates were also classified as low, appropriate, or high according to recommended dosing guidelines [10,11].

### 4.3. Statistical Analysis

For descriptive statistics, the variables were summarized in accordance with their type and distribution: normal variables as mean (standard deviation), non-normal variables as median (interquartile range), and categorical/binary variables as proportion (%). To assess associations between the use of higher importance antimicrobials with wound severity, state, and year, an additive crossed random effects multivariable ordinal logistic regression model was fitted with these variables trialed as fixed effects; the remaining explanatory variables were also trialed. As a secondary focus of the study was to determine if the clinic or veterinarian had the greatest variance in antimicrobial choice, additive crossed random effects for the clinics, veterinarians, and dogs were trialed, using Laplacian approximation, with the variance components assessed for being statistically significantly different to zero. In this sample, dogs were nested within clinics, as they could not be tracked between clinics; veterinarians were able to be tracked crossing between clinics. The odds ratios given in the model are multivariate adjusted and are for being above a specified antimicrobial importance level compared with being at or below that importance level.

For model validation, a population averaged model was fit, with robust variances for clustering on the clinic and veterinarian; the coefficients for the fixed effects were substantially similar to the crossed random effects model. A Wald test of the population-averaged proportional odds model versus the multinomial logit model, both clustered on clinic and veterinarian, was used to evaluate the global proportional odds assumption; there was no evidence the assumption does not hold (χ^2^ (6) = 9.1, *p* = 0.17).

The EPRs for the patients who did not receive antimicrobials were excluded as this resulted in violation of the proportional odds assumption. The EPRs for the DBW events missing a veterinarian identification code were also excluded, to allow for accurate modelling of the random effects. To evaluate representativeness of the model dataset, the full and analytic datasets were assessed for statistically significant differences using *t*-tests, Wilcoxon rank-sum (Mann–Whitney) test, and chi-square test for normal continuous, skewed continuous, and categorical variables respectively, with Bonferroni’s correction for multiple comparisons.

All statistical analyses were conducted in Stata version 16.1b and significance level was set at 0.05, except for violation of the proportional odds assumption, where it was 0.1. The Reporting of studies Conducted using Observational Routinely-collected health Data (RECORD) guidelines were used in the reporting of this study [35].

## 5. Conclusions

This study found that amoxicillin–clavulanic acid is the most commonly prescribed antimicrobial for DBW across Australia, and C&S testing is very rarely performed. There was evidence of increasing use of high importance antimicrobials over the past two decades. This is of concern as the lack of C&S testing and the increasing use of antimicrobials of high importance is contrary to antimicrobial prescribing guidelines and the principles of antimicrobial stewardship. The large variation between veterinarian prescription behaviors also suggests the need for targeted education on the prudent use of antimicrobials. The data collected in this study can be used as a baseline to track future trends in antimicrobial drug usage and adherence to prescription guidelines of Australian veterinarians treating DBW.

## Figures and Tables

**Figure 1 antibiotics-11-00055-f001:**
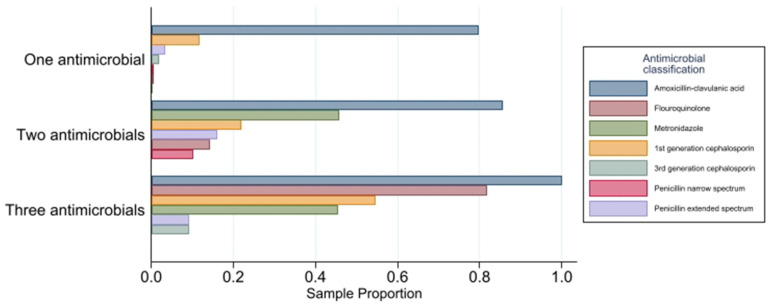
Sample proportions of antimicrobials by the number of oral and parenteral antimicrobials prescribed (omitted one dog with four antimicrobials dispensed).

**Figure 2 antibiotics-11-00055-f002:**
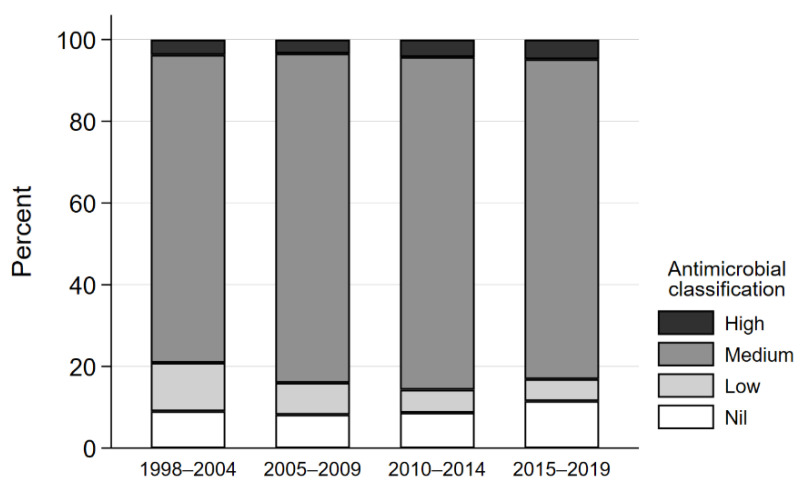
Antimicrobial importance prescribed per year category for the treatment of dogs presenting with dog-to-dog bite wounds from 1998–2004 to 2015–2018.

**Figure 3 antibiotics-11-00055-f003:**
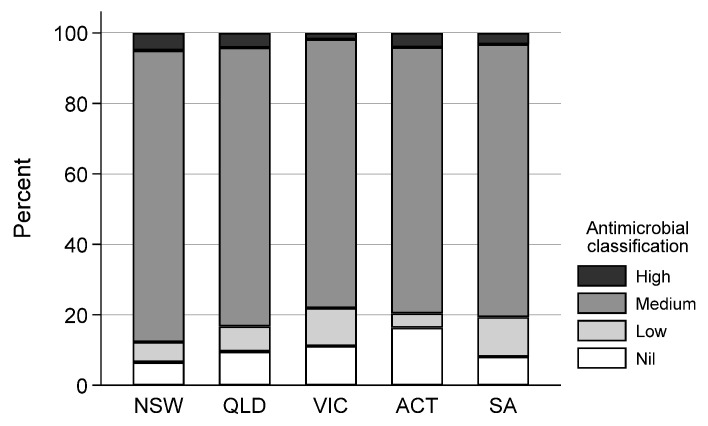
Antimicrobial importance prescribed per state (excluding WA due to low numbers) for the treatment of dogs presenting with dog-to-dog bite wounds from 1998 to 2018.

**Figure 4 antibiotics-11-00055-f004:**
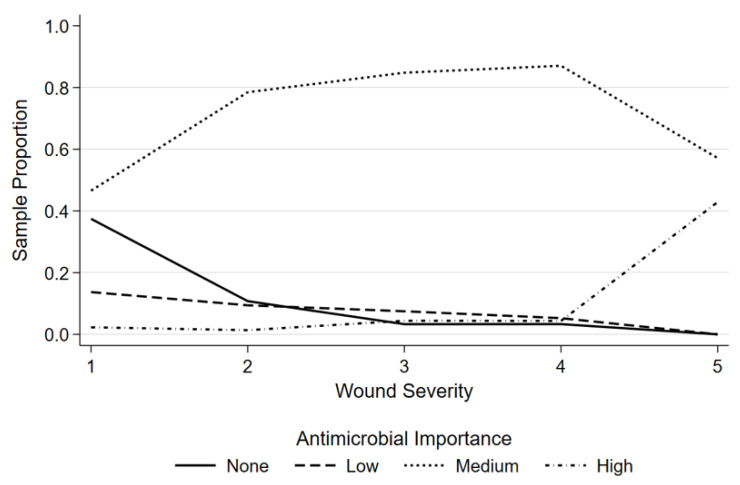
Antimicrobial importance class prescribed versus wound severity grade for 1713 dog-to-dog bite events presenting for treatment between 1998 to 2018.

**Figure 5 antibiotics-11-00055-f005:**
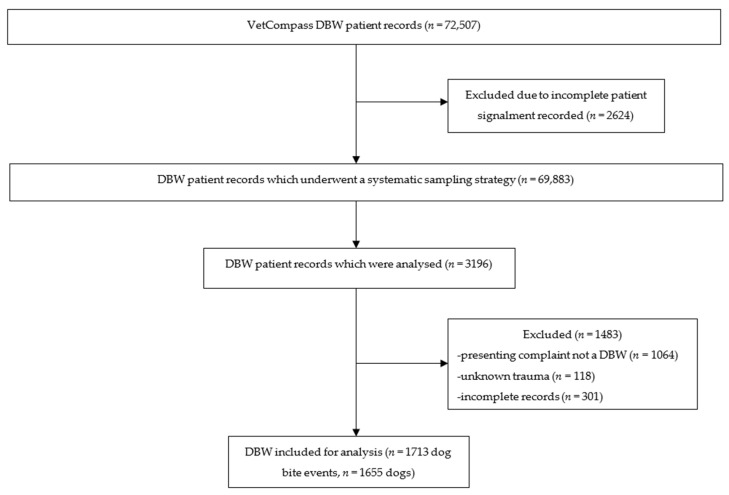
Flow diagram of the methodology of data collection for analysis of patients presenting for dog-to-dog bite wounds in the VetCompass Australia database from 1998-2018.

**Table 1 antibiotics-11-00055-t001:** Comparison of baseline patient characteristics of initial versus analytic (removal of dogs that did not receive antimicrobials and incomplete records) datasets for dogs presenting with dog-to-dog bite wounds from 1998 to 2018.

	Full Dataset (*n* = 1655 Dogs, 1713 DBW Events)	Analytic Dataset (*n* = 1375 Dogs, 1405 DBW Events)	*p* Value ^a^	*p* Value ^b^
**Median age, years** **(IQR)**	4.6 (2.0 to 8.0)	4.7 (2.0–8.0)	0.62	1.0
**Sex**			0.03 *	0.33
Male	901 (54.4%)	746 (53.1%)		
Female	754 (45.6%)	659 (46.9%)		
**Neuter**			0.58	1.0
Desexed	1125 (68.0%)	972 (69.2%)		
Entire	530 (32.0%)	433 (30.8%)		
**Median weight, kg** **(IQR)**	19.0 (8.4 to 26.2)	19.0 (9.0 to 26.2)	0.18	1.0
**Wound Severity ^c^**			<0.001 *	<0.001 *
Grade 1	219 (12.8%)	128 (9.1%)		
Grade 2	223 (13.0%)	179 (12.7%)		
Grade 3	457 (26.7%)	401 (28.5%)		
Grade 4	799 (46.6%)	689 (49.0%)		
Grade 5	15 (0.9%)	8 (0.6%)		
**State**			0.004 *	0.052
NSW	382 (22.3%)	337 (24.0%)		
QLD	993 (58.0%)	794 (56.5%)		
VIC	224 (13.1%)	177 (12.6%)		
ACT	49 (2.9%)	39 (2.8%)		
SA	62 (3.6%)	55 (3.9%)		
WA	3 (0.2%)	3 (0.2%)		
**Year ^c^**			<0.001 *	<0.001 *
1998–2004	321 (18.7%)	221 (15.7%)		
2005–2009	501 (29.3%)	395 (28.1%)		
2010–2014	518 (30.2%)	459 (32.7%)		
2015–2018	373 (21.8%)	330 (23.5%)		
**Number of antimicrobials ^c^**			<0.001 *	<0.001 *
No antimicrobials	158 (9.2%)	0 (0%)		
Monotherapy	1296 (75.7%)	1167 (83.1%)		
Polytherapy	259 (15.1%)	238 (16.9%)		

**Antimicrobial Importance**			<0.001 *	<0.001 *
No antimicrobials	158 (9.2%)	0 (0%)		
Low	126 (7.4%)	108 (7.7%)		
Medium	1360 (79.4%)	1231 (87.6%)		
High	69 (4.0%)	66 (4.7%)		
**C&S performed ^c^**	1 (0.1%)	1 (0.1%)	1.0	1.0
**Time of attack ^c^**			0.94	1.0
<1 h	68 (4.0%)	53 (3.8%)		
1–8 h	187 (10.9%)	154 (11.0%)		
8–24 h	234 (13.7%)	193 (13.7%)		
>24 h	622 (36.3%)	507 (36.1%)		
Unknown	602 (35.1%)	498 (35.4%)		
**Season of attack ^c^**			0.65	1.0
Spring	414 (24.2%)	335 (23.8%)		
Summer	409 (23.9%)	345 (24.6%)		
Autumn	404 (23.6%)	326 (23.2%)		
Winter	486 (28.4%)	399 (28.4%)		
**Median duration of hospitalisation (days)** **(IQR) ^c^**	0 (0 to 0)	0 (0 to 1)	0.18	1.0

**^a^** Uncorrected *p* value; **^b^**
*p* value corrected for multiple comparisons using Bonferroni method; **^c^** By unique consultation. IQR—interquartile range; * Significant at the 0.05 level; Queensland (QLD), New South Wales (NSW), Victoria (VIC), South Australia (SA), Australian Capital Territory (ACT), and Western Australia (WA).

**Table 2 antibiotics-11-00055-t002:** Number of events prescribed antimicrobials and route of administration in dogs which presented for treatment of dog-to-dog bite wounds from 1998 to 2018.

Antimicrobial	Number of Events	%
Amoxicillin-clavulanic acid parenteral (SQ)	578	33.7
Amoxicillin-clavulanic acid oral	1117	65.2
Cephalosporin (1st generation) parenteral (IV)	39	2.2
Cephalosporin (1st generation) oral	185	10.7
Cephalosporin (3rd generation) parenteral (SQ)	27	1.5
Fluoroquinolone parenteral (SQ, IV)	36	2.1
Fluoroquinolone oral	27	1.5
Metronidazole parenteral (IV)	8	0.4
Metronidazole oral	80	4.6
Penicillin narrow spectrum parenteral	25	1.4
Penicillin extended spectrum parenteral (SQ, IV)	38	2.2
Penicillin extended spectrum oral *	44	2.5
Sulphonamide parenteral (SQ)	2	0.1
Trimethoprim Sulphonamide oral	5	0.2
Tetracycline oral	7	0.4
Lincosamide oral	24	1.4
Topical antimicrobial *	147	8.5

* Neomycin. Subcutaneous (SQ), intravenous (IV).

**Table 3 antibiotics-11-00055-t003:** Antimicrobial dosage, frequency, and duration of administration prescribed to treat dog-to-dog bite wounds in 1713 dogs, which presented for treatment between 1998 and 2018.

	Dosage (mg/kg)			Frequency (Dose per Day)			Duration (Days)		
	Median	Minimum	Maximum	Median	Minimum	Maximum	Median	Minimum	Maximum
Amoxicillin-clavulanic acid parenteral	9.1	2.8	43.8	1	1	2	1	1	5
Amoxicillin-clavulanic acid oral	14.1	1.2	48.1	2	2	3	7	2	21
1st generation cephalosporin parenteral	14.5	2.4	25	4	1	5	1	1	5
1st generation cephalosporin oral	18.7	3.5	36.3	2	1	3	7	5	21
3rd generation cephalosporin parenteral	8	0.1	11.9	1	1	1	21	21	21
Fluoroquinolone parenteral	5	0.1	23.7	1	1	1	1	1	1
Fluoroquinolone oral	5.8	2.2	12.5	1	1	2	7	2	14
Metronidazole parenteral	10	5	19.2	1.5	1	2	1	1	2
Metronidazole oral	16.35	6.9	34.9	2	1	2	7	3	14
Penicillin narrow spectrum parenteral	16.7	8.6	38.6	1	1	1	1	1	5
Penicillin extended spectrum parenteral	13.3	4.4	26.8	1	1	4	2	1	7
Penicillin extended spectrum oral	11	5.3	24.5	2	1	3	6	4	14
Sulphonamide parenteral	12.9	11.4	14.4	1	1	1	1	1	1
Sulphonamide oral	13.6	2.4	29.8	2	2	2	6	5	10
Tetracycline oral	3.95	2.5	50	2	1	2	10	7	21
Lincosamide oral	6.1	2.3	13.4	2	2	2	7	4	14

**Table 4 antibiotics-11-00055-t004:** Antimicrobial dosage [10,11] for each class of antimicrobial importance prescribed to dogs that presented for treatment of dog-to-dog bite wounds from 1998 to 2018.

Antimicrobial Classification	Antimicrobial Dosage	Number of DBW Events	%
Low	Low	37	29.3
	Appropriate	12	9.6
	High	2	1.6
	Not enough data	75	59.5
Medium	Low	400	29.4
	Appropriate	546	40.1
	High	323	23.7
	Not enough data	91	6.7
High	Low	30	43.4
	Appropriate	25	36.2
	High	14	20.2
	Not enough data	0	0

**Table 5 antibiotics-11-00055-t005:** Crossed random effects multivariable ordinal multivariable logistic regression results for prescribing an antimicrobial of higher importance for dog-to-dog bite wounds presenting for treatment from 1998 to 2018 (*n* = 1405 consults, 1375 dogs, 610 veterinarians, 120 clinics).

	Odds Ratio	95% Confidence Interval	*p* Value
**Wound severity grade**	1.75	1.43–2.13	<0.001 *
**Year of Consultation** 1998–2004	Baseline		0.007 *
2005–2009	1.70	0.93, 3.11	0.09
2010–2014	2.37	1.26, 4.45	0.007 *
2015–2018	3.23	1.63, 6.42	0.001 *
**Surgery**			0.027 *
No surgery performed or recommended	Baseline		
Surgery performed	1.75	1.11, 2.78	0.017
Surgery recommended, not performed	0.57	0.19, 1.76	0.33
Cutpoint 1	−0.92	−1.59, −0.24	
Cutpoint 2	6.72	5.60, 7.84	
Clinic variance	0.54	0.21, 1.38	<0.001 *^a^
Veterinarian variance	2.14	0.95, 4.82	<0.001 *^a^

* Significant at the 0.05 level. The *p* values for the individual levels of the categorical variables compared to the baseline are given, with a bolded *p* value for all levels of that categorical variable combined. ^a^ Likelihood ratio test of variance component.

**Table 6 antibiotics-11-00055-t006:** Antimicrobial classification adapted from the Australian Strategic and Technical Advisory Group on Antimicrobial Resistance relevant to the treatment of DBW [34].

Antimicrobial Importance	Antimicrobial
Low	penicillin
	sulphonamides
	tetracyclines neomycin (topical antimicrobial)
Medium	amoxicillin-clavulanic acid
	1st & 2nd generation cephalosporins
	lincosamides
	metronidazole
	piperacillin/tazobactam piperacillin/clavulanic acid
High	3rd generation cephalosporins
	fluoroquinolones

## Data Availability

This research was undertaken with the assistance of information and other resources from the VetCompass Australia consortium under the project “VetCompass Australia: Big Data and Real-time Surveillance for Veterinary Science” (McGreevy et al., 2017), which is supported by the Australian Government through the Australian Research Council LIEF scheme (LE160100026).

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
