# Peer review of "A VetCompass Australia Study of Antimicrobial Use in Dog-to-Dog Bite Wounds (1998–2018)"

_antibiotics, 2022, doi:10.3390/antibiotics11010055_

Round 1

Reviewer 1 Report

In this study, the authors conducted a comprehensive retrospective study entitled "A VetCompass Australia study of antimicrobial use in dog-to-dog bite wounds (1998-2018)". The authors analyzed the VetCompass Australia database to describe the antibacterial prescribing habits of Australian clinics for dog-to-dog bite situations. Evaluate signs, culture and susceptibility testing (C&S) results, antimicrobial treatment, geographic location, and outcomes based on 1,713 dog bites. The authors found that antimicrobials were prescribed in 86.1%. Among various antibiotic options, an amoxicillin-clavulanic acid was prescribed in 70% of cases. In addition, it was confirmed that antimicrobial has high importance in wound severity and surgery. The subject of the authors' retrospective study is timely and very interesting. IIt will also benefit many clinical veterinarians as it has been proven based on more than 1500 cases over 20 years. However, if the following focus is added, it will be a more interesting and impactful study.

1. The graph in Figure 2 is displayed as difficult to understand. It seems that it needs to be presented as a bar graph or a more understandable format.
2. The authors' research topics are mainly focused on clinical significance/outcome. It would be good if the side effects related to antibiotic resistance, including antibiotic abuse and MRSA, which are currently a problem in human clinical cases, could be further reinforced.
3. In addition, as the authors mentioned, it would be better if there was an analysis of the results of an antibiotic susceptibility test or a bacterial culture test on a wound. As a reader and reviewer of this paper, I recommend a discussion on the necessity of these antibiotic susceptibility tests and bacterial culture tests.

Author Response

  1. The graph in Figure 2 is displayed as difficult to understand. It seems that it needs to be presented as a bar graph or a more understandable format.  Thank you for your suggestion, we have re-done the figure into a bar graph.

  2. The authors' research topics are mainly focused on clinical significance/outcome. It would be good if the side effects related to antibiotic resistance, including antibiotic abuse and MRSA, which are currently a problem in human clinical cases, could be further reinforced. Thank you for your suggestion, we have added the following paragraph in the discussion: “The extent to which antimicrobial resistance is affecting the health of humans and animals is not fully known. There are concerns that emerging resistance among bacteria could escalate, with unpredictable efficacy of antimicrobials , and some bacterial infections could become untreatable [26]. There is now increasing global pressure to develop strategies to protect the effectiveness of existing and new antimicrobials by reducing selection pressure driving emerging resistance in bacteria [26, 28]. Methicillin-resistant Staphylococcus aureus (MRSA) is a significant human pathogen in hospitals and commu-nities. MRSA is now recognized as an emerging pathogen of animals. Resistant infections might result from treatment with inappropriate antimicrobials [29]. As these pathogens are considered to be transmissible between humans and animal’s they pose a genuine threat not only to hospitalized animal patients but also to the work place safety of veterinary personnel and animal owners [30]. Veterinarians should therefore critically evaluate current antimicrobial usage to identify ways to prevent MRSA infections such as hand washing, masks, and use of topical disinfectants in an attempt to improve the health and wellbeing of animals and humans.” (Lines 603-617).

  3. In addition, as the authors mentioned, it would be better if there was an analysis of the results of an antibiotic susceptibility test or a bacterial culture test on a wound. As a reader and reviewer of this paper, I recommend a discussion on the necessity of these antibiotic susceptibility tests and bacterial culture tests. Thank you for your recommendation. We have added the following in the discussion “Obtaining wound cultures for organism identification and antimicrobial susceptibility testing is considered gold standard for the treatment of any wound, not just DBW. Culture and susceptibility testing should ideally be considered routine practice for any wound, not only reserved for severe cases or treatment failure [24, 25]. Submission of appropriate specimens for bacterial culture, pathogen identification and susceptibility testing allows an evidence-based approach for drug selection [26]. It can, however, increase the costs associated with patient care, although other costs associated with improper drug selection (prolonged treatment, use of expensive antimicrobials and increased morbidity and mortality) can outweigh the financial costs of testing [26].” (Lines 589-597).

Reviewer 2 Report

Dear Authors, 

I carefully read your manuscript and I found it simply amazing. 

I would like to suggest only a small correction at L389 and to move Figure 1 in Material and methods section (I think that it would be more appropriate considering that it is useful to understand the selection method). 

Author Response

I would like to suggest only a small correction at L389 and to move Figure 1 in Material and methods section (I think that it would be more appropriate considering that it is useful to understand the selection method). Thank you for your suggestion, we have moved Figure 1 to the materials and methods section and corrected the spelling mistake on line 389.

Reviewer 3 Report

This manuscript is very well-written and provides meaningful insights into the topic

Author Response

No comments or recommended suggestions from Reviewer 3.